

# Effects of adults aging on word encoding in reading Chinese: evidence from disappearing text

Zhifang Liu[1], Yun Pan[2], Wen Tong[3] and Nina Liu[4]

[1] Department of Psychology, Ningbo University, Ningbo, Zhejiang, China
[2] Department of Psychology, Guizhou Normal University, Guiyang, Guizhou, China
[3] Department of Psychology, Shanxi Normal University, Linfen, Shanxi, China
[4] Academy of Psychology and Behavior, Tianjin Normal University, Tianjin, Tianjin, China

## ABSTRACT

The effect of aging on the process of word encoding for fixated words and words presented to the right of the fixation point during the reading of sentences in Chinese was investigated with two disappearing text experiments. The results of Experiment 1 showed that only the 40-ms onset disappearance of word $n$ disrupted young adults' reading performance. However, for old readers, the disappearance of word $n$ caused disruptions until the onset time was 120 ms. The results of Experiment 2 showed that the disappearance of word $n + 1$ did not cause disruptions to young adults, but these conditions made old readers spend more time reading a sentence compared to the normal display condition. These results indicated a reliable aging effect on the process of word encoding when reading Chinese, and that the encoding process in the preview frame was more susceptible to normal aging compared to that in the fixation frame. We propose that sensory, cognitive, and specific factors related to the Chinese language are important contributors to these age-related differences.

# INTRODUCTION

Reading is an important practiced skill for daily life. This skill is acquired in early life and remains remarkably stable in adult age. However, many studies have recently revealed some subtle age differences in reading; that is, old adults spend more time to comprehend texts, make longer fixations, and make more regressions than young adults (*Kemper, Crow & Kemtes, 2004*; *Kliegl et al., 2004*; *Stine-Morrow, Miller & Herzog, 2006*; *Rayner et al., 2006*; *Laubrock, Kliegl & Engbert, 2006*; *Rayner, Castelhano & Yang, 2009*). Changes in optics and neural transmission that occur with normal aging often lead old adults to experience a range of subtle visual deficits (*Owsley, 2011*), which may contribute substantially to their decline in reading ability. *Rayner et al. (2006)* and *Rayner, Castelhano & Yang (2009)* also proposed that these age-related differences are largely attributed to the sensory and cognitive decline associated with normal aging. However, the nature of this decline has yet to be fully determined. Of particular importance for the present study is that the aging effect on visual word encoding may be related to both the changes in visual sensory

Corresponding author
Zhifang Liu, lzhf2008@163.com

abilities and the cognitive decline associated with normal aging. Investigation of how visual word encoding changes with age will inform future developments of computational models for eye movement control, such as the E-Z Reader and SWIFT models, which have been shown to successfully simulate some of the aging differences in word processing for alphabetic languages (*Laubrock, Kliegl & Engbert, 2006*; *Rayner et al., 2006*; *Reichle, Rayner & Pollatsek, 2003*; *Engbert et al., 2005*). However, there are no models that have attempted to simulate aging changes when encoding Chinese words during reading.

The disappearing text paradigm, in which the fixated word $n$ or word $n+1$ disappears after a certain fixation duration (i.e., 40, 60, or 80 ms), has been the most effective method to resolve the issue of visual word encoding. With this paradigm, researchers have found that most of the visual information of the fixated word necessary for reading can be acquired within the initial 50–60 ms during a fixation for young readers of English, which means that they are able to encode all the needed visual information of a fixated word within this time frame (*Rayner et al., 1981*; *Liversedge et al., 2004*; *Blythe et al., 2009*; *Blythe et al., 2011*). Perhaps the most striking result was that even when the word disappeared after a fixation of 60 ms, a robust frequency effect on the target words remained, which meant that word processing influenced eye movements in reading, which is consistent with the cognitive-control models of eye movements (*Rayner et al., 2003*). *Rayner, Liversedge & White (2006)* also investigated the visual encoding of word $n+1$ with disappearance experiments and found that the disappearance of word $n+1$, after word $n$ was fixated for 60 ms, greatly impaired reading, which indicated not only the importance of visual coding for word $n+1$, but also that the time needed for encoding words in parafoveal vision is longer than for words in foveal vision when reading English. It should be noted that the E-Z Reader model can account not only for young adults' eye movement data for word $n$ disappearing text but also for their performance in word $n+1$ disappearing conditions during reading of English (*Pollatsek, Reichle & Rayner, 2006*; *Reichle et al., 2009*).

Clinical impairments are commonly found in the population of old adults, and these impairments have profound effects on reading performance (*Paterson, McGowan & Jordan, 2013*). Further evidence has revealed that reading performance decreases significantly with age even in people whose visual acuity is good (*Lott et al., 2001*). Moreover, given the nature of sensory and cognitive decline in old adults, it is likely that aging affects encoding of visual word information. Thus, it is safe to hypothesize that old adults need more time to encode words during reading. However, previous research on this topic in reading English has revealed that for old readers, the disappearance of the fixated word caused relatively greater reading slowdown compared to young adults, even in the 60-ms onset disappearing conditions. Due to the absence of an interaction between age group and disappearing onset when the control condition was removed, and the word frequency effect for both groups in the disappearing conditions, the authors argued that the effect of disappearing onset is comparable in old and young readers of English text (*Rayner et al., 2010*). Thus, these results indicated no age effects on visual word encoding when reading English. However, it has been an open question whether aging effects on visual word encoding are a language-specific phenomenon or whether they are universal and apply to other languages such as Chinese. Furthermore, it is important to explore the aging pattern

of word encoding in parafoveal vision, as preview processing of words is common for both young and adult readers (*Rayner, Castelhano & Yang, 2009*; *Rayner, Castelhano & Yang, 2010*). However, research on this topic is still rare both in reading English and Chinese.

Encoding words of alphabetic languages during reading involves the discrimination of one-dimensional, linear combinations of letters or phonological units and then encapsulate them into more permanent representations (*Reichle et al., 2009*). However, encoding Chinese words involves the recognition of the two-dimensional, pattern-like structure of characters, which engages unique mental processes (*Zhang et al., 2012*). Thus, comparing aging effects on visual word encoding of Chinese to those of English is extremely valuable for developing computational models that simulate aging differences during reading. As mentioned above, with the disappearing text paradigm, researchers have extensively investigated the time needed for encapsulating words in foveal and parafoveal vision (*Rayner et al., 1981*; *Rayner et al., 2003*; *Rayner et al., 2006*; *Liversedge et al., 2004*) and the developmental and aging issues of encapsulating the visual information of a fixated word (*Blythe et al., 2009*; *Blythe et al., 2011*; *Rayner, Castelhano & Yang, 2010*). As previous studies have revealed, aging does not have an effect on encoding visual word information in the fovea during reading English (*Rayner, Castelhano & Yang, 2010*). In contrast to the well-documented effects of disappearing text and the effects of aging on reading English, however, empirical studies concerning disappearing text in reading Chinese have been relatively sparse and are still in their infancy. Only one study using the disappearing text paradigm was found—in which visual information of the fixated word necessary for sentence comprehension can be acquired within the initial 50–60 ms for young adult readers of Chinese—which indicated that they could encapsulate the fixated visual word information as quickly as young adult readers of English (*Liu, Zhang & Zhao, 2011*). Although researchers have investigated whether word difficulty factors (e.g., usage and visual complexity) affected young and old adults differently (*Wang et al., 2016*; *Zang et al., 2016*), there has been no study, as far as we know, that manipulated the exposure time of words to directly examine the aging effect on Chinese visual word encoding in reading.

In view of the substantial differences between the two script types, findings on word encoding of alphabetic scripts (such as English) during reading cannot be directly extended to reading Chinese. Firstly, Chinese writing is logographic whose written style is completely different from that of alphabetic text. Evidence has shown that the processes underlying lexical identification of Chinese words are very different from those of alphabetic languages (*Zhou & Marslen-Wilson, 2000*). The prevalent assumption has been that Chinese lexical identification is a form-to-meaning process with little involvement of phonology; therefore, it appears that orthography dominates over phonology, and that orthographic encoding is a core process in Chinese word identification (*Perfetti, Liu & Tan, 2005*). *Zhang et al. (2012)* also posited that Chinese is a more thoroughly visual language compared to alphabetic scripts, and thus emphasizes the role of visual processing in word recognition. Secondly, Chinese is a language with no spaces to separate words in a text, and texts written in Chinese are formed by strings of box-like symbols (i.e., characters). Although series of studies conducted by Bai and colleagues have suggested the importance of words (opposed to characters) for reading and learning Chinese, which are the same as for alphabetic

languages (*Bai et al., 2008*; *Blythe et al., 2012*; *Shen et al., 2012*; *Bai et al., 2013*), character processing is the necessary stage for multi-character word recognition (*Li, Rayner & Cave, 2009*; *Shen & Li, 2012*; *Li et al., 2014*). Therefore, readers must segment character strings into words during preview (*Yan et al., 2010*; *Shu et al., 2011*; *Yan et al., 2015*; *Gu & Li, 2015*), which may lead to greater difficulty for old people when encoding visual word information. With the characteristics of Chinese text mentioned above, aging effects on visual encoding of words during reading may be larger than those during reading English text. Thus, in this study, we employed two disappearing text experiments with longer interval onset times compared to previous research conducted by *Rayner, Castelhano & Yang (2010)* to explore the effect of aging on the time needed for encoding words both in foveal and parafoveal vision (i.e., word $n$ and $n+1$) during reading of Chinese text.

The primary goal of the current study, therefore, was to examine age-related changes in visual word encoding. If such changes could be established, how much time would old adults need to encode the visual word information in foveal and parafoveal vision as well as young adult readers of Chinese? We further examined aging effects on the time needed for encoding not only the fixated word (word $n$) but also the word to the right of fixation (word $n+1$) when reading Chinese by two disappearing text experiments, respectively. The first experiment was conducted to explore aging effects on encoding word $n$, in which the onset times of the word $n$ disappearance were manipulated. In the second experiment, the same onset times of word $n+1$ disappearance were manipulated, and by doing so, we intended to clarify the aging effects on encoding word $n+1$ and compare it to the aging effects on encoding word $n$. The onset times of the disappearance manipulation in both experiments were 40, 80, 120, and 160 ms, with a 40-ms interval of the disappearing manipulation. The logic of both experiments was straightforward. If the onset time were not sufficient for encoding before the word disappeared, normal reading performance would be impaired under this onset time of disappearing text. Moreover, if young and old adults had different requirements on the time needed for encoding the words to read normally, these differences should be revealed by the effectiveness of each onset times of disappearing manipulation to sustain normal reading performance in each age group.

## GENERAL METHOD

### Human ethics

The data were anonymously analyzed. The subjects provided verbal and written informed consent by signing a form to receive money for their participation. The current study was approved by the ethics committee of the Department of Psychology of Ningbo University (approval number: 20150901).

### Participants

Sixty adults from the University of Ningbo and local community participated in the experiments. Of these, 15 young adults ($M = 20.5$ years, range 18–22 years) and 15 old adults ($M = 66.7$ years, range 60–73 years) participated in Experiment 1, and another 15 young adults ($M = 20.4$ years, range 18–22 years) and 15 old adults ($M = 66.8$ years, range 60–73 years) participated in Experiment 2. The young and old adult groups did not differ

in number of years of schooling (15.8 years for the young adults and 15.4 years for the old readers). All participants were right-handed with normal or corrected-to-normal vision. Subjects with eyesight problems were asked to wear their glasses before they participated in the experiment, and there was no group difference in corrected vision scores by the Tumbling E acuity chart (old adults: $M = 4.99$, $SE = 0.15$; young adults: $M = 4.97$, $SE = 0.13$; $t = 1.117$, $p > 0.05$). All participants reported that they could clearly see the text in the no disappearing condition after the practice phase in the experiment. They were all native Chinese speakers and were paid ¥30 for participation. None of them was aware of the purpose of the experiment or had previously participated in other similar experiments.

## Apparatus

The participants' movements of the right eye were recorded with an Eye Link 1,000 device manufactured by SR Research Ltd. The eye tracker is an infrared video-based tracking system, and its camera samples the pupil location and pupil size at a rate of 1,000 Hz. This system also has high spatial resolution (<0.01° RMS). The sentence stimuli were presented on a 19-inch DELL LCD monitor with a 1024 × 768 pixel resolution and refresh rate of 75 Hz. The sentences were displayed in Song font, and the size of each Chinese character was 28 × 28 pixels subtending approximately 0.63° visual angle. The distance between the participant and screen was 75 cm.

## Design and stimuli

Both experiments followed a 2 (group: young adults *vs.* old adults) × 5 (disappearing onset: no disappearing, 40 ms, 80 ms, 120 ms, 160 ms) mixed design. The former variable was a between-subjects factor and the latter a within-subjects factor. Experiment 1 explored whether disappearing word *n* influenced young and old Chinese readers' reading behavior, and Experiment 2 explored whether disappearing word *n* + 1 influenced young and old Chinese readers' reading behavior. Both groups of participants read the Chinese sentences in a normal condition (control) and four experimental conditions in which the word *n* or word *n* + 1 disappeared after the designated interval (40, 80, 120, or 160 ms). An immediate re-fixation on word *n* did not result in its re-appearance until the reader made an eye movement to a new word in the disappearing manipulations. Figure 1 shows an example of the disappearing manipulations in detail; when the reader fixated on word "垃圾," which was word *n*, it disappeared after 40, 80, 120, or 160 ms and was not presented until the reader made an eye movement to a new word. When the reader made an eye movement and fixated on word "通道," the disappeared word "垃圾" reappeared; and, after 40, 80, 120, or 160 ms, "通道" disappeared (note: the sentences all contained seven or eight two-character words).

Figure 2 shows an example of the disappearing manipulations in detail; when the reader fixated on word "垃圾," which was word *n*, the word "通道," which was word *n* + 1, disappeared after 40, 80, 120, or 160 ms and was not presented again until the reader made an eye movement to a new word. When the reader made an eye movement and fixated on word '通道," the disappeared word re-appeared, and after 40, 80, 120, or 160 ms, "需要" disappeared.

| | | | |
|---|---|---|---|
| （a） | 垃圾通道需要及时清理。 | | [beginning of fixation] |
| | * | | |
| （b） | 通道需要及时清理。 | | [after40/80/120/160ms] |
| | * | | |
| （c） | 垃圾通道需要及时清理。 | | [a new fixation] |
| | * | | |
| （d） | 垃圾  需要及时清理。 | | [after40/80/120/160ms] |
| | * | | |
| （e） | 垃圾  需要及时清理。 | | [An immediate re-fixation did't result word reappearance] |
| | * | | |

**Figure 1  Examples of the word *n* disappearance conditions.** The asterisk indicates a fixation location.

| | | | |
|---|---|---|---|
| （a） | 垃圾通道需要及时清理。 | | [beginning of fixation] |
| | * | | |
| （b） | 垃圾  需要及时清理。 | | [after40/80/120/160ms] |
| | * | | |
| （c） | 垃圾通道需要及时清理。 | | [a new fixation] |
| | * | | |
| （d） | 垃圾通道  及时清理。 | | [after40/80/120/160ms] |
| | * | | |
| （e） | 垃圾通道  及时清理。 | | [An immediate re-fixation did't result word reappearance] |
| | * | | |

**Figure 2  Examples of the four word *n* + 1 disappearance conditions.** The asterisk indicates a fixation location.

It is important to note that the refresh rate of the screen can affect the precise timing of a display. The disappearing text manipulations were initiated when the eye fixated on word n. The refresh rate of the screen was 75 Hz in the present study, therefore, in each disappearing condition, there was a potential additional 15 ms of delay before the words disappeared after the eye moved to word *n*. Thus, the actual onset times of the disappearing manipulations in both experiments (Experiments 1 and 2) were 55, 95, 135, or 175 ms.

Eighty sentences were used as stimuli in the experiments. These sentences all contained seven or eight two-character words. Considering that 72% of Chinese words are two-character words—when word tokens are taken into account, 27% words are two-character words—it was easy to make up fluent Chinese sentences with only two-character words (*Li et al., 2012*). Ten college students, who did not participate in the experiments, were asked to rate the difficulty of these sentences on a 7-point scale (e.g., a score of 7 was "very difficult"). The resulting mean difficulty score was 1.65. Another 10 college students, who did not participate in the experiments, were asked to rate the naturalness of these sentences on a 7-point scale (e.g., a score of 7 was "very natural"). The resulting mean naturalness score was 6.30. Each sentence was shown in one of five display conditions. All eighty sentences were randomized and sampled using a Latin square, so that each participant saw sixteen sentences shown in each of the five display conditions: a control and four disappearing display conditions. By doing this, we wanted to ensure that all sentences were

shown equally often in the five display conditions and prevent repetition of any sentence for each participant. Sentences were shown to each participant in a randomized order across two sessions.

There were 12 additional sentences (five of which had questions) for practice in the first session. The second session was the formal experiment. To confirm that participants were reading the sentences carefully, there were 27 filler sentences that were randomly inserted throughout the block. A Yes/No comprehension question was presented at the end of each filler sentence, and the participants were asked to answer the questions by manually pressing the buttons.

## Procedure

Participants were tested individually. Before the experiment, they were informed that they would read sentences presented in different disappearing manners. The participants were instructed to read and understand the sentences and were asked to push a button box to terminate the current display when a sentence was completed. A comprehension task occasionally appeared after a sentence, and the participants were asked to answer a Yes/No question by pressing different buttons. A chin rest was inserted to ensure that the participants' heads remained still. A calibration procedure was executed prior to the beginning of the experiment, in which the participant was instructed to look at each of three fixation points arranged along a horizontal line across the center of the screen. Once the eye tracker had been calibrated with satisfactory accuracy (mean error less than 0.5°), the sentence was presented. The eye tracker was re-calibrated before the next trial when the error from drift correction of the current trial was greater than 0.5°. There were 4–6 re-calibrations for each subject. The experiment was approximately 25 min in length.

## Data analysis

Since the purpose of this study was to investigate aging effects on visual word encoding by checking the differences in time needed for word encoding between young and old adults, we compared each disappearing condition to the control separately in young and old adults. One word-independent measure, reading time, was the most important reference measure with the disappearing text paradigm (*Blythe et al., 2009*); three word-dependent measures, such as mean gaze duration (mean gaze duration on all the words in a sentence), probability of skipped words in the initial pass when reading a sentence, and probability of regressed words during reading were analyzed as supplementary data. The reason to adopt the reading time measure as the main reference dependent variable is that readers may trade off fixation duration and number during disappearing text reading (*Rayner et al., 1981*; *Liversedge et al., 2004*; *Liu, Zhang & Zhao, 2011*), which in turn could lead the word-dependent measures not being sensitive to the word encoding process. All eye movement data above or below three standard deviations from the mean were excluded; as a result, 3.7% of the data in total were removed prior to conducting the analyses. Data were analyzed by linear mixed models (LMM) using the lme4.0 package (*Bates et al., 2016*) and the latest version of R, which was retrieved from https://www.r-project.org/. The full models included random intercepts for participants and stimuli and included additional

random slopes in which participants and stimuli were allowed to vary with respect to the fixed effects. If the model did not converge, we sequentially simplified the maximal model until it converged. The models were run on log-transformed reading time and mean gaze duration. Logistic LMM models were used for analyzing the probability of skipped words and probability of regressed words. The contrasts of the main effects were defined with sliding contrasts in the MASS package (*Venables & Ripley, 2002*). The last version of this package was also retrieved from https://www.r-project.org/. We report regression coefficients ($b$), standard errors ($SE$), and $t$ values ($t = b/SE$), regression coefficients approximately twice as large as their $SE$ (abs[$t$ or $z$ value] > 2) are interpreted as significant at the 5% level.

The rationale of the disappearing text paradigm is to increase the amount of visual information available per fixation and to test the minimum amount of time for encoding the disappeared word to maintain a normal reading behavior when no viewing restriction is applied. The goal of the present study was also to investigate how young and old readers of Chinese differ in the time needed for encoding the visual word information into a more stable code. Therefore, we consider the treatment contrast between young and old groups to be particularly well suited for the goal of the present study. Inferential statistics were based on a prior treatment contrast with the no disappearing condition as the reference category for the four disappearing text conditions. Then, the group differences when reading the text without disappearing conditions from the two experiments were explored. Furthermore, older adults' reading data from the two experiments were compared to explore differences of aging between encoding word $n$ and word $n+1$. In this analysis, experiment (Experiment 1 vs. Experiment 2) was specified as a fixed effect, and there were four contrasts (contrast 1: no disappearing vs. 40-ms disappearing onset, contrast 2: no disappearing vs. 80-ms disappearing onset, contrast 3: no disappearing vs. 120-ms disappearing onset, contrast 4: no disappearing vs. 160-ms disappearing onset). The model provided statistics with respect to five main effects (experiment, contrast 1, contrast 2, contrast 3, and contrast 4) and four interactions (experiment and contrast 1, experiment and contrast 2, experiment and contrast 3, experiment and contrast 4).

## EXPERIMENT 1

Experiment 1 investigated the differences in time needed for encoding the visual information of the fixated word into a more stable code between young and old adults. We presented the text either entirely as normal or in the disappearing text mode, in which word $n$ disappeared after the designated interval (40, 80, 120, or 160 ms) from the time it was fixated.

### Results

The mean accuracy of all participants for the 27 comprehension sentences exceeded 80%, and there were no significant effects of display condition and age group ($ps > 0.10$). Namely, the comprehension levels were high in all situations, which indicated that participants read and fully understood the sentences. All the data are summarized in Table 1.
**Table 1** **The mean and standard deviations of the measures across conditions and age groups in Experiment 1.** Means and standard deviations are computed across subjects' means. The standard deviations are given in parentheses.

|  |  | RT | MeanGazeDur | Pro.Skip(%) | NO.Reg |
|---|---|---|---|---|---|
| Young adults | Control | 2654(816) | 254(38) | 24.9(15.4) | 1.7(1.1) |
|  | 40 ms | 2761(822) | 273(46) | 25.3(15.6) | 2.1(1.1) |
|  | 80 ms | 2738(848) | 274(56) | 25.0(15.1) | 1.9(0.9) |
|  | 120 ms | 2625(840) | 268(51) | 25.5(16.9) | 1.8(1.0) |
|  | 160 ms | 2728(813) | 269(49) | 25.5(17.3) | 1.7(1.0) |
| Aged adults | Control | 3334(1001) | 312(71) | 13.5(8.3) | 2.1(0.9) |
|  | 40 ms | 3815(819) | 288(55) | 18.1(8.8) | 3.5(0.9) |
|  | 80 ms | 3632(835) | 310(61) | 16.5(10.1) | 3.0(1.0) |
|  | 120 ms | 3389(816) | 304(60) | 15.4(10.2) | 2.7(1.0) |
|  | 160 ms | 3439(835) | 308(68) | 16.6(9.4) | 2.6(1.0) |

**Notes.**

Control, the normal display condition; RT, sentence reading time in millisecond; MeanGazeDur, mean gaze duration in milliseconds; Pro.Sikp, probability of words skipped in the initial pass reading; NO.Reg, number of regressions.

### Reading time

Only the 40-ms disappearing onset condition prolonged young adults' sentence reading time compared to the control, $b = 0.040$, $SE = 0.015$, $t = 2.623$; while the other disappearing manipulations did not interrupt their normal reading performance, $bs < 0.022$, $SEs > 0.012$, $ts < 1.60$. However, in old adults, the 40-ms and 80-ms disappearing onset conditions differed from the control (40-ms onset vs. control: $b = 0.062$, $SE = 0.019$, $t = 3.235$, 80-ms onset vs. control, $b = 0.041$, $SE = 0.015$, $t = 2.797$), with the shorter disappearing onset producing longer reading times. The 120-ms and 160-ms onset disappearing conditions did not interrupt older adults' reading performance, $bs < 0.019$, $SEs > 0.013$, $ts < 0.35$.

### Mean gaze duration

The disappearing manipulations influenced young and old adults differently. All the disappearing onset conditions prolonged the young adults' gaze duration compared to the no disappearing condition, $bs > 0.018$, $SEs < 0.01$, $ts > 2.12$; with longer onset times, the mean gaze duration was shorter. However, in old adults, the 40-ms disappearing onset condition led to shorter gaze duration compared to the no disappearing condition, $b = -0.031$, $SE = 0.015$, $t = -2.110$. There were no differences between the other disappearing conditions and the control, abs[$b$ values] $< 0.011$, $SEs > 0.008$, abs[$t$ values] $< 0.992$, that is, the longer disappearing onsets did not affect mean gaze duration.

### Probability of skipped words

The word $n$ disappearing manipulations affected the skip probability of young and old adults differently. All disappearing conditions did not differ from the no disappearing condition for young adults, $bs < 0.036$, $SEs > 0.097$, $Zs < 0.97$; however, the 40-ms and 160-ms disappearing conditions led old adults to skip words more often compared to the no disappearing condition, $bs > 0.285$, $SEs < 0.133$, $Zs > 2.195$.

### Probability of regressed words

Compared to the control, none of the disappearing conditions influenced the regression probability of words in young adults, abs[$b$ values] < 0.07, $SEs$ > 0.10, abs[$Z$ values] < 0.571. However, the 40-ms, 80-ms, and 160-ms disappearing conditions made old adults regress more often than the control condition, $bs$ > 0.245, $SEs$ < 0.156, $Zs$ > 2.059, with shorter disappearing onset increasing the probability of regressed words.

## Discussion

The most important aspect of the sentence reading time data is that old adults were much more disrupted by the disappearance of the fixated word than were the young adults, and that the more immediate disappearing onset was more disruptive to text processing. In particular, we found that the disappearance of word $n$ influenced reading in the young and aged groups differently, that is, only the 40-ms onset of the fixated word disappearing condition interrupted the reading time in the young adults, which suggested that the time needed for young Chinese readers to encode the visual information of word $n$ for normal reading is about 55–95 ms. Thus, the results from this experiment replicated prior findings both in English and Chinese (*Rayner et al., 1981*; *Liversedge et al., 2004*; *Liu, Zhang & Zhao, 2011*). However, for old adults, the disappearance of word $n$ prolonged the reading time until the disappearing onset was 135 ms (120 ms disappearing onset). That is, the period needed for visual encoding of the fixated word was 95–135 ms for old adults, which is inconsistent with findings on reading English, in which the authors concluded that the disappearing onset effect was comparable in old and young English readers (*Rayner, Castelhano & Yang, 2010*).

The word-dependent eye movement measures provided additional evidence for the effects of normal aging on the time needed for visual encoding of fixated words. In particular, young and old adults adopted a completely different oculomotor strategy to read disappearing text. As seen from Table 1 and Fig. 3, young adults gazed at word of interests longer, skipped more words, and made slightly more regressions during disappearing text reading. Although the disappearing onset effects on the probability of skipped and regressed words were not reliable, the higher skip probability traded off the longer mean gaze duration, which in turn blunted the negative influence of the disappearing manipulations on sentence reading time. In contrast, the disappearing manipulations led old adults to gaze at words for shorter times and make more skips in the first pass of reading; the disappearing manipulation also made old people regress more often compared to the no disappearing condition. It is easy to understand that more regressions caused a longer reading time. The differences between old and young adults manifested when reading in the word $n$ disappearing conditions. Based on the reading time data, these results suggested that it was insufficient for old adults to visually encode the fixated word within the 95-ms display of visual information (i.e., 80-ms onset disappearance of word $n$ still disrupted old adults' reading performance), so that they had to comprehend the text by recognizing the word on the right side of fixation (i.e., word $n+1$). These findings confirmed that compared to the young adults, old readers need more time to encode the visual information of the fixated word.

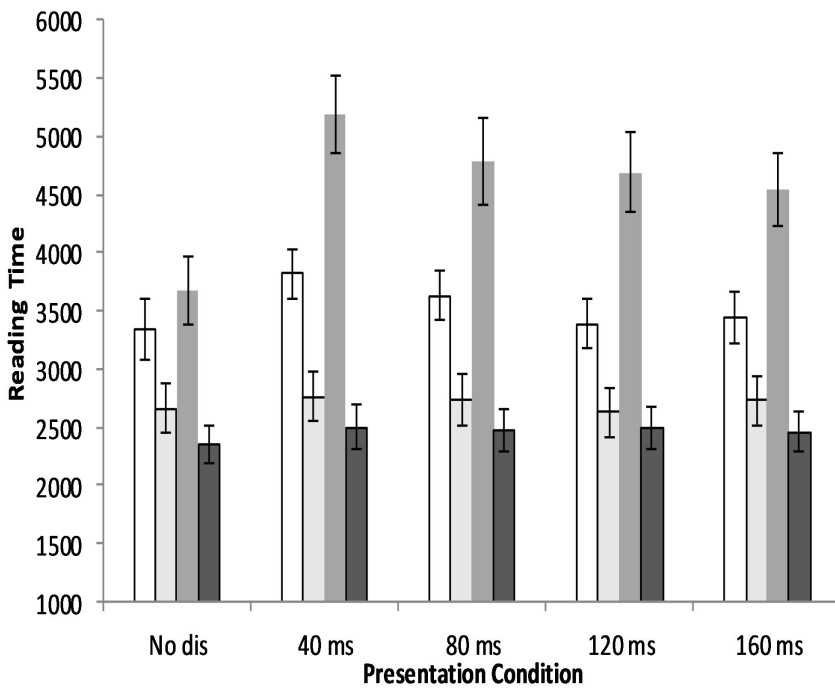

**Figure 3** **Sentences reading time of young and old adults for each of the five presentation conditions in two experiments.** Error bars show the standard error each group under each condition. In each group of the bars, the first bar was the data of old adults when reading word $n$ disappearing text, the second was the data of young adults when reading word $n$ disappearing text, the third was the data of old adults when reading word $n+1$ disappearing text, and the fourth was the data of young adults when reading word $n+1$ disappearing text.

# EXPERIMENT 2

Experiment 2 investigated the differences in time needed for encoding the visual information of word $n+1$ for normal reading performance between young and old adults. We either presented the text entirely as normal or presented it in the disappearing paradigm in which word $n+1$ disappeared after the designated interval (40, 80, 120, or 160 ms) from when word $n$ was fixated.

## Results

The mean accuracy of all participants for the comprehension sentences exceeded 80%. There were no reliable effects for the display conditions and age groups ($ps > 0.10$). All eye movement data above or below three standard deviations from the mean were excluded, and 2.9% of the data in total were removed prior to conducting the analyses. The dependent measures are summarized in Table 2.

### Reading time

Similar to Experiment 1, we contrasted all the disappearing conditions with the no disappearing condition for young and old adults separately, and found that the word $n+1$ disappearing manipulations affected young and old readers differently. That is, compared to the no disappearing condition, none of the disappearing manipulations interrupted the

**Table 2 The mean and standard deviations of the measures across conditions and age groups in Experiment 2.** Means and standard deviations are computed across subjects' means. The standard deviations are given in parentheses.

|  |  | RT | MeanGazeDur | Pro.Skip(%) | NO.Reg |
|---|---|---|---|---|---|
| | Control | 2347(640) | 262(31) | 21.1(9.4) | 1.4(0.7) |
| | 40 ms | 2498(762) | 293(57) | 26.8(9.4) | 1.5(0.8) |
| Young adults | 80 ms | 2475(723) | 287(42) | 26.8(9.2) | 1.6(0.8) |
| | 120 ms | 2494(691) | 284(42) | 24.8(8.7) | 1.5(0.8) |
| | 160 ms | 2460(700) | 279(39) | 24.7(8.5) | 1.5(0.8) |
| | Control | 3669(1123) | 351(108) | 7.6(4.6) | 1.9(0.9) |
| | 40 ms | 5195(1299) | 378(67) | 14.3 (4.7) | 4.4(2.1) |
| Aged adults | 80 ms | 4784(1440) | 396(80) | 14.0(6.0) | 3.6(1.7) |
| | 120 ms | 4687(1331) | 396(75) | 13.0(6.5) | 3.3(1.6) |
| | 160 ms | 4544(1214) | 391(83) | 12.8(6.1) | 3.0(1.2) |

**Notes.**

Control, the normal display condition; RT, sentence reading time in millisecond; MeanGazeDur, mean gaze duration in milliseconds; Pro.Sikp, probability of words skipped in the initial pass reading; NO.Reg, number of regressions.

young adults' normal reading performance, $bs < 0.023$, $SEs > 0.014$, $ts < 1.41$. However, all the disappearing manipulations interrupted the old readers' normal reading performance, $bs > 0.092$, $SEs < 0.023$, $ts > 4.516$, with shorter disappearing onset producing longer reading times.

### Mean gaze duration
Compared to the no disappearing condition, all the disappearing conditions prolonged the young adults' gaze duration on the word of interest, $bs > 0.024$, $SEs < 0.015$, $ts > 2.265$. The 80-ms, 120-ms, and 160-ms disappearing conditions also prolonged the old adults' gaze duration, $bs > 0.053$, $SEs < 0.020$, $ts > 3.115$, but the 40-ms disappearing condition did not influence old adults' mean gaze duration, $b = 0.041$, $SE = 0.023$, $t = 1.73$.

### Probability of skipped words
All the disappearing conditions increased the young adults' skip probability, $bs > 0.244$, $SEs < 0.089$, $Zs > 2.788$. In old adults, all the disappearing conditions increased skip probability compared to the control, $bs > 0.628$, $SEs < 0.181$, $Zs > 3.65$, with the shorter disappearing onset producing higher skip probability.

### Probability of regressed words
Compared to the control, none of the disappearing conditions influenced the word regression probability in young adults, $bs < 0.163$, $SEs > 0.126$, $Zs < 1.467$. However, all the word $n+1$ disappearing manipulations caused old adults to regress more often, $bs > 0.618$, $SEs < 0.165$, $Zs > 3.961$, with the shorter disappearing onset producing more regressions.

### Discussion
The aging effect on the process of encoding word $n+1$ when reading Chinese was explored in this experiment. It was again confirmed that old adults were much more disrupted by the disappearance manipulations than were the young adults, and that the more immediate

disappearing onset was more disruptive to processing. As described in the previous section, none of the word $n+1$ disappearance manipulations impaired the normal reading performance of young adults. However, for the old adults, the disappearing manipulations prolonged sentence reading time, with the shortest disappearing onset time (40 ms) causing the greatest interruptions. The results of the young adults are inconsistent with previous studies in English, which indicated that the 60-ms onset time disappearance of word $n+1$ impaired young readers' normal reading performance (*Rayner et al., 2006*). The results on old adults indicated that it becomes more difficult to encode word $n+1$ within the limited time, and even 175 ms might not be sufficient (the 160-ms onset time of word $n+1$ disappearing condition still impaired the old adults' reading performance). Thus, these data indicated that old adults' reading suffered more seriously when encoding word $n+1$ than when encoding the fixated word.

The word-dependent eye movement measures provided additional evidence to support the conclusions made from the reading time data. It was found that readers also adopted a strategy of trading off gaze duration, regressions, and skips to read the sentences treated by the word $n+1$ disappearing manipulations. As seen in Table 2, young adults adopted a totally different oculomotor strategy to read word $n+1$ disappearing text compared to the old readers. In particular, the disappearance of word $n+1$ affected the mean gaze duration for young adults and affected the probability of skipped words almost equally for both young and old adults. The longer mean gaze duration during the word $n+1$ disappearing manipulations were traded off with a higher skip probability for both groups, that is, when reading word $n+1$ disappearing text, readers were inclined to fixate fewer words with longer reading time. It is quite likely that the trade-off between fixation number and time occurred similarly in this experiment, which led to longer gaze duration in the disappearing text reading for both groups; however, the reduced preview caused by the word $n+1$ disappearing text may have also contributed to the prolonged gaze duration of old readers. The measure of regressions further revealed the group differences. It is crucial to illustrate the differences between young and old adults in reading time and regressions, as seen from Table 2; none of the disappearance onsets of word $n+1$ brought on more regressions for the young adults but brought on more regression for old adults.

## COMPARISON BETWEEN EXPERIMENTS

It should be pointed out that across the two experiments, the control conditions differed very little and any variability could be due to the between-participant manipulation, since different subjects participated in the two experiments. In order to better understand the age-related differences in performance when reading Chinese, a series of analyses were conducted to compare the differences between young and old adults in the no-disappearing conditions from the two experiments. As evident from these analyses, when reading the text under normal display conditions, old adults took more time than young adults (young adults: $M = 2502$ ms, old adults: $M = 3513$ ms; $b = 0.186$, $SE = 0.049$, $t = 3.765$), had longer gaze duration (young adults: $M = 256$ ms, old adults: $M = 331$ ms; $b = 0.185$, $SE = 0.049$, $t = 3.756$), and tended to make more regressions than young adults

(young adults: $M = 0.24$, old adults: $M = 0.32$; $b = 0.551$, $SE = 0.221$, $t = 2.489$), which is consistent with results from research on aging effects of alphabetic script reading (*Kemper, Crow & Kemtes, 2004*; *Kliegl et al., 2004*; *Rayner et al., 2006*; *Rayner, Castelhano & Yang, 2009*). However, old Chinese readers skipped fewer words than young readers (young adults: $M = 0.233$, old adults: $M = 0.107$; $b = -1.007$, $SE = 0.213$, $t = -4.723$). This finding differs from aging effects on reading English, which suggested that compared to old readers of English (*Rayner, Castelhano & Yang, 2009*), old Chinese adults adopted a more cautious oculomotor strategy when reading Chinese, and they were more inclined to look at the words one-by-one during reading. These findings are valuable to understand the aging effects on word encoding when reading Chinese.

In order to better understand the different aging effects on word encoding between words in foveal and parafoveal locations, the old adults' reading data in both experiments were further compared. In this analysis, only the reading time was considered, because old adults adopted two completely different oculomotor strategies to read in the disappearing text conditions of Experiment 1 and 2 (as seen in Tables 1 and 2, respectively), which may have caused the word-dependent measures not being susceptible to the different aging effects on encoding word $n$ and word $n+1$. The results showed that old adults spent more time to read the sentences in Experiment 2 than to read the sentences in Experiment 1, $b = 0.113$, $SE = 0.037$, $t = 3.006$. All the contrasts were reliable, $bs > 0.056$, $SEs < 0.008$, $ts > 7.224$, and so were the interactions, $bs > 0.07$, $SEs < 0.02$, $ts > 4.513$, which indicated that the disappearance of word $n+1$ impaired old adults' reading performance more seriously than the disappearance of word $n$. This indicated that the aging effect on encoding word $n+1$ was larger than that on encoding word $n$.

## GENERAL DISCUSSION

With two disappearing text experiments, the aging effect on visual word encoding (word $n$ and $n+1$) was investigated during reading Chinese. The present study provides evidence concerning the word encoding process of older Chinese readers. Generally, it was found that compared with young adults, old adults read more slowly, gaze at words for longer, and make more regressions (see Tables 1 and 2, Figs. 3 and 4, and section on comparison between experiments). This adds to the growing evidence that old readers suffer from greater reading difficulties (*Kemper, Crow & Kemtes, 2004*; *Kliegl et al., 2004*; *Stine-Morrow, Miller & Herzog, 2006*; *Rayner et al., 2006*; *Rayner, Castelhano & Yang, 2009*; *Laubrock, Kliegl & Engbert, 2006*). We also found some age changes that are specific of Chinese, that is, compared to reading alphabetic languages such as English and German, old adults reading Chinese employ a more cautious eye movement strategy (i.e., they skip words less frequently), which is consistent with previous studies (*Liu et al., 2015*; *Wang et al., 2016*; *Zang et al., 2016*). These researchers proposed that increased difficulty in processing word boundary in the preview phase with age causes older adult readers to use a more careful reading strategy to compensate for this difficulty. However, only a few studies have directly explored the aging effect on parafoveal processing of words. The disappearing text paradigm to investigate aging effects on processing of visual information of word $n$ and $n+1$ is particularly well suited to assess this issue.

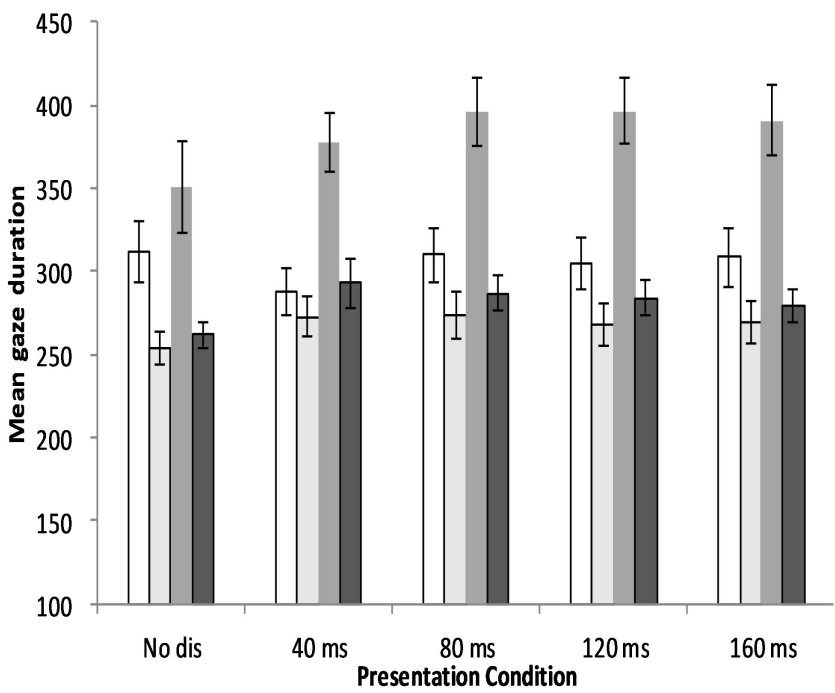

**Figure 4** **Mean gaze duration of young and old adults for each of the five presentation conditions in the two experiments.** Conventions as for Fig. 3.

Although previous studies have shown that young adult readers of Chinese need about the same time to encode the fixated word as readers of alphabetic languages, it does not mean that both languages share the same pattern of normal aging in word encoding during reading. The differences in time needed for encoding word *n* between young and old adults during reading Chinese were investigated in Experiment 1. The results indicated that young adults can encode the visual information of a fixated word within 55–95 ms display time (i.e., 80-ms onset disappearance of word *n* did not disrupt young adults' reading performance), which is generally consistent with prior research both for reading English and Chinese, suggesting that young adults can read fairly well when they see the fixated word for about 55 ms before it disappears (*Rayner et al., 2003*; *Liversedge et al., 2004*). However, old Chinese adults need more time to encode the visual information of word *n* (i.e., 95–135 ms), which is not consistent with findings from old readers of English (*Rayner, Castelhano & Yang, 2010*). Comparing the results of this experiment to those obtained by Rayner and colleagues (*Rayner, Castelhano & Yang, 2010*), confirms that aging has a more serious impact on encoding fixated words when reading Chinese. We propose two possible factors for this difference. The first is that Chinese readers may rely more heavily on the preview process than English readers. The second is that word identification of Chinese text is a form-to-meaning process with more emphasis on visual processing (*Zhou & Marslen-Wilson, 2000*; *Perfetti, Liu & Tan, 2005*; *Zhang et al., 2012*). Both factors may also interact with the visual decline (especially the disproportionate decline in peripheral vision); thus, making the encoding of visual word information during

reading Chinese more susceptible to normal aging. The second experiment was conducted to check the first factor directly.

With the word $n+1$ disappearing text manipulations, we explored the aging effects on the process of encoding word $n+1$ in Experiment 2. *Rayner et al. (2006)* observed that the 60-ms onset disappearance of word $n+1$ impaired young adults' reading performance of English text equally compared to its immediate disappearance. According to *Pollatsek, Reichle & Rayner (2006)*, attention plays a key role in encapsulating the visual information into a more permanent representation. By this account, 60 ms were sufficient for encoding the fixated word but not to identify the fixated word and then shift attention to word $n+1$ in time for it to be encoded. Thus, this is inconsistent with the attention gradient model, which assumes that multiple words are encoded in parallel because, if attention is allocated to both word $n$ and word $n+1$, there should be sufficient time to convert the visual information of both words into an orthographic code, enabling lexical processing of either word to continue without disruption after they disappear (*Reichle et al., 2009*). However, as seen from the results and discussion section for Experiment 2, none of the word $n+1$ disappearing conditions disrupted young Chinese adults' reading time (see Table 2 and Fig. 3). Word skip probability was traded off by longer mean gaze duration in the disappearing manipulations compared with the control condition in young adults (see Table 2 and Fig. 4). Thus, these results indicated that they could encode the visual information of word $n$ and $n+1$ into a relatively stable code in parallel. Given that young adults still needed some time to encode the visual word information within a fixation frame (as seen from Experiment 1), it is safe to conclude that young Chinese adult readers could encode the characters in word $n$ and $n+1$ successfully within 55 ms after they first fixated on word $n$. (If the output of encoding word $n+1$ is word-based, the reader should have no need to re-encode it when it is in the fixation frame.) Thus, the results of young adults when reading word $n+1$ disappearing text is consistent with the notion proposed by Li and his colleagues who argued that Chinese readers process characters in parallel during reading (*Li, Rayner & Cave, 2009*; *Ma, Li & Rayner, 2015*).

The results of Experiment 2 suggest that young adult readers of Chinese are more inclined to encode the text in parallel than young adult readers of English, which also indicates that Chinese readers rely on the preview process. The influence of the word $n+1$ disappearing manipulations on old adults' reading is also informative for developing computational models of eye movement control when reading Chinese. All the word $n+1$ disappearing manipulations interrupted old adults' reading performance, in that they exhibited longer gaze duration and made more regressions when reading disappearing text than when reading no disappearing text (see Table 2 and Fig. 4). Thus, both the word-independent and word-dependent variables revealed that 175 ms were not sufficient to encode word $n+1$ for old readers. Comparing the old adults' reading time measure in Experiment 1 to that in Experiment 2 confirmed that the disappearing onset of word $n+1$ interrupted reading more seriously than the same disappearing onset of word n, which meant that old adults needed more time to encode word $n+1$ compared to word n. Old adults' reading suffered more when they encoded the text of word $n+1$ than when they encoded the fixated text, which confirmed that Chinese readers may rely more heavily on the preview process

than English readers. Previous studies claimed that old people have difficulty segmenting words in the preview phase, which leads them to adopt a more careful reading strategy (*Wang et al., 2016*; *Zang et al., 2016*). Although the results of the present experiments do not refute this deduction, it can be concluded that the lower efficiency in encoding the characters of the previewed word was also an important contributor to the more cautious eye movement strategy adopted by old adults, which was a specific age change in reading Chinese.

The reasons why old Chinese readers had greater difficulty encoding the visual information of the previewed word than of the fixated word consisted of sensory, cognitive, and factors specific to Chinese; all of these factors are not mutually exclusive contributors to this difference. We propose that the sensory factor may be the most important contributor to this aging effect when reading words in the two locations. As a previous study has shown, visual acuity decreases with normal aging disproportionately for peripheral vision relative to regions that are closer to the fixation location (*Cerella, 1985*). *Paterson, McGowan & Jordan (2013)* also found that old adults gradually lose the ability to process detailed visual information in both foveal and parafoveal regions and rely much more on coarse-scale components and a much wider region of text when reading compared to young adults. Given that the perceptual span and preview word segmentation when reading Chinese was mediated by visual factors (e.g., font size of characters), which are specific to Chinese (*Yan et al., 2015*), the deficits in parafoveal vision may lead to a smaller perceptual span for old readers. Thus, together with the previous finding, it is safe to conclude that deficits in parafoveal vision might be an important reason for the lower preview efficiency of old readers. The cognitive reason is attention-related. The lack of disruption from the manipulation of word $n+1$ disappearance for young adults indicated that this group could encode the visual information of both words (word $n$ and $n+1$) in parallel. Old readers adopted a more conservative eye movement strategy to compensate for their lower preview efficiency, and the longest disappearing onset time of word $n+1$ still impaired reading performance, which indicated that old readers of Chinese may identify the characters during reading more serially than the young readers. However, this issue needs to be examined in further studies.

The Chinese-specific reasons were as follows. Firstly, recognition of characters is necessary for the identification of multi-character words, and Chinese character recognition is relatively independent from the processing of the word they belong to *Li, Rayner & Cave (2009)* and *Shen & Li (2012)*. This means that Chinese readers process information at multiple levels when they recognize the words composed of multiple characters. Secondly, evidence has shown that Chinese word identification involves a form-to-meaning process that is totally different from alphabetic languages (*Zhou & Marslen-Wilson, 2000*; *Perfetti, Liu & Tan, 2005*). A previous study also found that Chinese word encoding puts more emphasis on visual processing, but the recognition of Chinese multi-character word orthography is not achieved until 200 ms after presentation (*Zhang et al., 2012*). Thus, the processes for encoding visual information from Chinese script into more stable representations are expected to be more susceptible to aging. Aging factors (i.e., perceptual speed and others) may also increase the difficulty transitioning from character recognition

to multi-character word identification (*Salthouse, 1994*). In sum, the current results demonstrate that as Chinese readers get older, they develop adaptive shifts into more cautious oculomotor strategies to compensate for their poorer word encoding ability in the preview during reading, which might not apply to other languages. The reasons for these age shifts may be largely due to the unique characteristics of Chinese. Although further research is needed on this issue, future studies on word encoding and eye movement control should emphasize not only universal characteristics but also those specific to the Chinese language.

### Funding
This study was sponsored by the K.C. Wong Magna Fund at Ningbo University. This work was also supported by the Humanities and Social Science Project from the Ministry of Education of China to Zhi-fang Liu (Grant No. 15YJC190014). The funders had no role in study design, data collection and analysis, decision to publish, or preparation of the manuscript.

### Grant Disclosures
The following grant information was disclosed by the authors:
K.C. Wong Magna Fund.
Humanities and Social Science Project from the Ministry of Education of China: 15YJC190014.

### Competing Interests
The authors declare there are no competing interests.

### Author Contributions
- Zhifang Liu conceived and designed the experiments, performed the experiments, analyzed the data, contributed reagents/materials/analysis tools, wrote the paper, prepared figures and/or tables, reviewed drafts of the paper, made sentences stimuli.
- Yun Pan performed the experiments, analyzed the data.
- Wen Tong contributed reagents/materials/analysis tools, wrote the paper, prepared figures and/or tables, reviewed drafts of the paper.
- Nina Liu contributed reagents/materials/analysis tools, reviewed drafts of the paper.

### Human Ethics
The following information was supplied relating to ethical approvals (i.e., approving body and any reference numbers):
Ethics committee of the Department of Psychology of Ningbo University: approval number: 20150901.

### Data Availability
The raw data has been supplied as Supplemental File.

## Supplemental Information

Supplemental information for this article can be found online at http://dx.doi.org/10.7717/peerj.2897#supplemental-information.

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
