# Peer review of "Effects of adults aging on word encoding in reading Chinese: evidence from disappearing text"

_PeerJ, doi:10.7717/peerj.2897_

## Round 0.1 · original submission · Major Revisions

· Academic Editor

Major Revisions

Dear Zhifang and colleagues,

I have now received two reviews from experts in the field and I would like to thank the reviewers for their careful consideration of the manuscript. The reviews raise a number of comments for your consideration. I believe addressing each of these points will help to clarify the work and increase its value and impact.

Please feel free to get in touch with me if you can have questions and I'll look forward to receiving an updated manuscript in the near future.

Best wishes,
Nic

·

Basic reporting

The article structure adheres to common scientific standards and introduces the necessary literature to follow the argumentation and to understand the motivation of the study. Figure and Tables are comprehensible, however I would appreciate an additional Figure. I would find it very helpful to see one of the dependent measures (e.g., gaze durations) as a function of the disappearing text onset times (for young and older adults and possibly for Experiment 1 and Experiment 2). As such, one could immediately see which conditions differ from each other and whether fixation durations reach an asymptotic level with a certain presentation duration.

Experimental design

The experimental design represents state-of-the-art research in eyetracking replicating disappearing text manipulations with Chinese readers that have been reported in studies published in other journals. I have to comments: (1) It would be beneficial to describe the calibration routines in more detail. For transparency, “Whenever necessary” (l. 167) should be explained further as the routines between labs differ somewhat and the reader should have the possibility to judge the data quality. This is particularly relevant for eyetracking experiments implementing gaze contingent text changes. (2) The authors are transparent about the fact that there were quite long delays in making the text disappear. This seems mainly originated in the untypically low monitor frequency of 75 Hz the authors used. In gaze-contingent setups one tries to get the highest refresh rates as possible (typically 150 Hz) to reduce such delays. For research I would recommend to use increased monitor frequency. (3) One reason for not having used a higher refresh rate may be that the authors used an LCD monitor instead of a CRT monitor. Generally, CRT monitors are preferred in experiments in which the timing on screen is important. If an LCD monitor was used this should be made explicit in the method section, and its implication should be discussed.

Validity of the findings

To my understanding, the authors have conducted an ANOVA only across subjects. In linguistic/reading research, this F1 analysis is typically compared with the complementary F2 analysis over items (here words). To avoid the problem of diverging results from the by-subject and the by-item analysis, linear mixed models are currently used as the optimal statistical method to control for the random effects of subjects and items.

Additional comments

I appreciated that the authors try to relate the results between the different dependent variables they analysed. This is often neglected in other studies. In Experiment 1, I would maybe emphasize the tradeoff between skips and regressions more strongly. There were more skippings but also more regressions, therefore resulting in a global increase of reading times. Interestingly, the skipping group effect is opposite to what is known from the literature. Typically, older adults skip more than younger adults. This may deserve further discussion.

Reviewer 2 ·

Basic reporting

No Comments.

Experimental design

No Comments.

Validity of the findings

No Comments.

Additional comments

Review of “Effects of adult aging on word encoding in Chinese reading: evidence from disappearing text” (#10145).

Using a classic disappearing text paradigm, the present study investigated the effect of aging on Chinese word encoding in foveal and parafoveal regions during reading. The results showed that 40 ms onset disappearance of word n (Expt 1) or word n+1 (after word n was fixated for 40ms, Expt 2) did not produce disruptions to young adult readers, however for older adult readers, even 80 ms onset disappearance of word n (Expt 1) or up to 160 ms onset disappearance of word n+1 (after word n was fixated for 160ms, Expt 2) still disrupted older adults’ reading performance, indicating that older adults experience more difficulty in word encoding during Chinese reading.

The question of whether older adult readers need more time to encode words from foveal and parafoveal vision during Chinese reading is interesting and may provide better understanding for the currently limited literature in this field. The experiments appear to be carefully conducted, and the authors provide detailed analyses for the data from within-experiments and between-experiments. I therefore feel that the paper could be suitable for publication in Peer J. However, I do have some questions and comments regarding the Introduction, Method and General Discussion sections which I would like to see addressed in a revision prior to publication.

Introduction
In the Introduction, the authors pointed out that there has been no research investigating aging effects on word encoding in reading Chinese. Actually there are at least three studies (Liu et al., 2015; Wang et al., 2016; Zang et al., 2016) examining word processing in young and older adult readers during Chinese reading, since the topics of these studies are so relevant to the current study, I would suggest extending this section by discussing more background research on aging effects during Chinese reading. Also, I think the theoretical motivation and hypotheses for the present study must be made stronger and clearer.

The authors mentioned that only one study used the disappearing text paradigm to investigate visual processing of the fixated word in Chinese reading, but this reference is missing, and it is not listed in the Reference section.

The onset times of disappearance in both experiments were 40, 80, 120, and 160ms (note the refresh rate of the screen was relatively low, 75 Hz – this itself is an issue which could affect the precise timings the authors are describing). I am wondering why the authors chose these particular onset times. And what hypothesis did the authors have for the young and older adults based on these timings. This aspect of the manuscript needs significant development.

Method
For the older adults’ reading in the experiments, it would be good if the authors tested readers’ visual acuity, for example, using a Tumbling E acuity chart, to make sure that they had normal or corrected to normal vision. If the authors did this testing, please add this information. If they did not, then this is a serious weakness in the manuscript as it is not clear whether the effects may be attributed to less efficient (linguistic) cognitive processing, or less effective visual processing.

Each experimental sentence was comprised of 7-8 two-character words, why? Please provide the naturalness ratings for the stimuli as normal Chinese sentences usually contain words with mixed lengths including one, two and three or more character words. This aspect of the experimental set up is not dealt with adequately in the paper as it currently stands.

The Design sections in Expt 1 and 2 are a bit repetitive. This is a stylistic suggestion, which the authors may choose to ignore, but I would suggest integrating the two Design sections to the General Method.

General Discussion
Much of the discussion in this section is repeated from the results section. I suggest limiting the discussion of results in the actual results section and instead expanding upon some important issues in the Discussion section. For example, the typical finding in English reading is that older adults are more likely than young adults to skip words and also tend to make more regressions to re-read text (Rayner et al., 2006, 2009), however in the present study it was found that older Chinese adults skipped less often compared to young adult readers. The authors should provide some explanation of why this might happen (see Zang et al., 2016).

The paragraph “The analyses on eye movement measures showed that…led to more skips and regressions” is hard to follow. The effect of disappearing onset was not reliable for the young adults for skipping probability (Table 1). Also, why did the authors argue that the disappearance of fixated words prolonged young adults’ gaze on words, which was traded off by the HIGHER skipping probability compared to the normal display condition? This section is not clear to me.

The authors argued that young adults could encode the visual information of both two-character word n and two-character word n+1 in parallel, it is not clear to me why this is consistent with Li et al.’s (2009) notion that readers process CHARACTERS in parallel? In fact in Li et al.’s model, characters are processed in parallel but words are processed serially. This inaccuracy should be dealt with.

The authors concluded that “deficits in parafoveal vision might be an important reason for the lower preview and text comprehension rate in old readers” (Line 489-490). First, how do the authors know that the problems lay with cognitive processing rather than simply visual processing if they did not test visual acuity. Second, it is not clear to me why the authors talked about the comprehension rates here, given that there were no differences between young and older adults in terms of their comprehension rates in both experiments.

Minor points:

Page 6, Line 66-67: “This is informative for revealing the age effect on visual word encoding in the foveal region phrase during reading”, Line 75: “…in foveal and parafoveal phrases”, the words “phrase” or “phrases” do not make sense.
Page 8, Line 144-145: “Each sentence was shown in one of five display conditions”, please specify what the five conditions are here.
Page 8, Line 148-149: “Sentences were shown to each participant in a randomized order across the two sessions”, the word “the” should be removed. Please check the other places to make sure the use of “the” is correct.
Page 11, Line 237, 245, and 254, “…is shown in Table 1” is redundant.
Page 13, Line 328 and 334, “… is shown in Table 2” is redundant.
Page 13, Line 325, “the 160ms onset time of word m + disappearing…” “m” does not make sense.
Page 16, Line 441, “Zhou, & Marslen-Wilson, 20001”, “20001”should be “2000”.

References:
Liu, P., Liu, D., Han, B., & Paterson, K. B. (2015). Aging and the optimal viewing position effect in Chinese. Front. Psychol.6:1656, doi: 10.3389/fpsyg.2015.01656.
Wang, J., Li, L., Li, S., Xie, F., Chang, M., Paterson, K.B., White, S.J., & McGowan, V. A. (2016). Adult age differences in eye movements during reading: The evidence from Chinese. Journals of Gerontology: Psychological Sciences, doi:10.1093/geronb/gbw036.
Zang, C., Zhang, M., Bai, X., Yan, G., Paterson, K.B., & Liversedge, S.P. (2016). Effects of word frequency and visual complexity on eye movements of young and older Chinese readers, The Quarterly Journal of Experimental Psychology, 69:7, 1409-1425, doi: 10.1080/17470218.2015. 1083594.

---

## Round 0.2 · Minor Revisions

· Academic Editor

Minor Revisions

Dear Zhifang,

Thank you for your very thorough consideration and response to the comments from the reviewers. I agree with the reviewers that the updates contribute to a more comprehensive piece of work that a valuable contribution to the literature.

As an aside, I wanted to add that the delay in feedback on the manuscript is entirely due to me being on leave. I would like to thank the reviewers for their timely and helpful consideration of the work. Please note that your contribution to science is being honoured next week in Peer Review Week!

In relation to the update, the reviewers raise some additional points for clarification which constitute minor revisions. Please consider these in a further update.

Best wishes,
Nic

·

Basic reporting

No Comments.

Experimental design

No Comments.

Validity of the findings

As I am not an expert on Chinese reading I will focus my comments on the methodological issues of the paper. I appreciated the change to linear mixed-effects models (LMMs). The first paragraph of the section “Data analysis” describes some model details, and the treatment contrasts seem sufficiently motivated. However, did the main model also contain the interaction between the group contrast and the disappearing onset contrasts?

I also miss some information when reporting the model results in the results section. The main effect of group with group and disappearing onset as treatment contrasts reflects that older adults read slower than younger adults in the no disappearing condition. This should not be confused with an average slower reading rate of older subjects across all disappearing conditions. The intercept of the present model is the average reading time of young adults in the no disappearing condition, not the grand mean.

The analyses that “contrasted all the disappearing manipulations with the no disappearing control condition for young and old adults” (p.9, ll.16-17), are this additional analyses on the age subsets of the data? This should be indicated, probably in the section “Data analysis”.

Fixation durations are commonly log-transformed before computing LMMs because they are often not normally distributed. This can affect the significance of fixed effects. It would be worth looking at the distribution of the dependent variables reading times and gaze durations in the present study and transform them if necessary. Skipping probability (if I understand correctly, this is here not a binomial variable) and regression number are possibly also not normally distributed (poisson?) and the model parameters should be checked after appropriate transformation.

Page 13, lines 14-15: I think the estimated slope (b) and the t-value should be negative.

Additional comments

Minor issues:
(1) page 5, line 28: change „onset time“ into „onset times“
(2) page 9, line 39: The “however” suggests that the skipping result for older adults stands in contrast to the results of younger adults but this seems to not be the case.
(3) page 10, line 41: I don’t think that “abandon” is not the appropriate word here. Indeed, the older adults were still able to understand the sentences as the high comprehension rates showed.
(4) page 13, lines 32-38: The contrasts should at this point be already clear to the reader. I would move this detailed explanation of the (basic) model to “Data analysis” and only briefly mention the changes here.
(5) page 14, line 15: Better “less frequent” instead of “infrequently”
(6) page 16, line 7-8: Here it seems better to repeat what the “first suggested factor” was instead of referring to an earlier part of the discussion.

Reviewer 2 ·

Basic reporting

No comments.

Experimental design

No comments.

Validity of the findings

No comments.

Additional comments

When I reviewed this manuscript in the first round, I was mainly concerned with the lack of background research on aging effects during Chinese reading, less clear theoretical motivation and hypotheses, the issues of onset times of disappearance and readers’ visual acuity test, and the problems with the discussion. In the revision the authors have tried to deal with these concerns and other additional minor comments. They also provided new analyses with R. I consider the current version has made a substantial improvement in the readability of the manuscript. I feel that this paper can be considered for publication after some corrections with data analyses and typographical amendments.

In the revision, the authors mentioned “Data were analyzed by linear mixed models (LMM) using the lme4.0 package (Bates, Maechler, & Bolker, 2012) within R by simultaneously taking participants and stimuli as crossed random effects.” However the authors did not indicate whether they computed random participants and stimuli slopes, as it is very important to include random slopes in the model (see Barr, Levy, Scheepers, & Tily. 2013).

In addition, the authors should indicate whether models were run on the log-transformed or raw fixation times, and whether a logistic lme model was applied for skipping rates.

The names “Hǎikiǒ” and “Hyǒnǎ” are not correct; please check carefully through the MS.

---

## Round 0.3 · Minor Revisions

· Academic Editor

Minor Revisions

Dear Zhifang and colleagues,

Many thanks for your updates to the manuscript. Once again I am grateful to the two reviewers for considering the updates and for the further comments on the work. I would be pleased if you could address the comments that Reviewer 1 (Sarah) has made with regards to the analysis, as well as the minor adjust suggested by Reviewer 2. I consider these to be minor revisions and I look forward to your revision.

Best wishes,
Nic

·

Basic reporting

Reporting of mixed-effects analyses

I reviewed this interesting paper already two times and was now asked to have a look at the revised method and results section, particularly at the changes relating to mixed-effects models. I list my comments below in the order they appeared to me during reading:

(1) Data Analysis (p.8, l.18): better “we compared each disappearing conditions” rather than “we compared the disappearing conditions”
(2) Data Analysis (p.8, l.25): reading time is the dependent variable, not the independent variable
(3) Data Analysis (p.8, l.27-28), also see (p.13, l.37-38): To my understanding, the tradeoff between fixation durations and fixation number leads to an insensitivity of the reading measure. In fact, the word-based fixation measures and probabilities are local measures viewed as reflecting the word-by-word adaptation of the oculomotor system towards processing difficulties. In contrast to that, the sentence reading times are a global measure and effects on the local level can, in principle, cancel each other out at the global level. Therefore, I would argue that the word-based measures are more sensitive for word processing.
(4) Data Analysis (p.8, l.32): I would recommend always to use the latest version of the lme4 package (and R in general). As a highly active open source tool, packages are frequently updated and important changes can occur from one version to the next.
(5) Data Analysis (p.8, l.33-34): I worry that the comment of the other reviewer was a bit misleading. Crossed random effects are always treated as random intercepts in (G)LMMs and represent the variance of the participants and stimuli with respect to the models’ fixed effect intercept. What Barr et al. (2013) recommend is to also include the random slopes for each fixed effect estimate, that is to allow the fixed effect to vary for each participant and each stimuli. Thus, there is no such thing than “random participants and stimuli slopes” in an (G)LMM. There are random intercepts for participants and stimuli and there can be additional random slopes in that participants (and/or stimuli) are allowed to vary with respect to the fixed effects. Here it would mean that each disappearing onset contrast would be submitted as random slope for subjects and stimuli. I am not sure whether the authors have really submitted random slopes to their models. I rather doubt it because to estimate the maximal model, this would mean a substantial increase in model parameters, and it is likely that, given the low number of data points, this model would not converge.
(6) Comparison Between Experiments (p.13, l.26): The minus signs should be preceding the number and not the letter.

Experimental design

No comments

Validity of the findings

No comments

Additional comments

My suggestion about how to deal with point 5 would be to report the theoretically interesting model (without random slopes) and to acknowledge (e.g., in a footnote) the recommendation of Barr et al. arguing that the full model could not be reported due to conversion problems. Alternatively, one could sequentially simplify the maximal model until it converges.

Reviewer 2 ·

Basic reporting

No comments.

Experimental design

No comments.

Validity of the findings

No comments.

Additional comments

I am happy with all the changes that the authors have made, and I feel that this paper can be considered for publication after a correction in Page 16, Line 431 - "-b" or "-t" is not accepted.

---

## Round 0.4 · accepted · Accept

· Academic Editor

Accept

Hi Zhifang,

Thank for you for the corrections to the manuscript so late in the year. I am very pleased to be able to accept your work for publication in PeerJ! And thank you for working through the series of revisions.

I did notice that Figures 3 and 4 are missing y-axes labels and I'd urge you to consider having a label for the x-axes as well (i.e., 'Presentation Condition') for clarity. I have accepted the article rather than requesting minor revisions for these - hopefully these can be updated as part of the production process rather than requiring a formal revision.

Best wishes,
Nic